# Utilisation of Potassium Chloride in the Production of White Brined Cheese: Artificial Neural Network Modeling and Kinetic Models for Predicting Brine and Cheese Properties during Storage

**DOI:** 10.3390/foods13193031

**Published:** 2024-09-24

**Authors:** Katarina Lisak Jakopović, Irena Barukčić Jurina, Nives Marušić Radovčić, Rajka Božanić, Ana Jurinjak Tušek

**Affiliations:** Department of Food Engineering, Faculty of Food Technology and Biotechnology, University of Zagreb, Pierottijeva 6, 10000 Zagreb, Croatia; katarina.lisak.jakopovic@pbf.unizg.hr (K.L.J.); nmarusic@pbf.hr (N.M.R.); rbozan@pbf.hr (R.B.); ana.tusek.jurinjak@pbf.unizg.hr (A.J.T.)

**Keywords:** white brined cheese, sodium replacement, potassium chloride, artificial neural network, physicochemical properties

## Abstract

Excessive sodium consumption is a worldwide problem, prompting the industry to develop sodium-reduced products and substitute salts. High sodium consumption is a significant risk factor for high blood pressure, cardiovascular disease and kidney disease. Excessive sodium intake also impairs the immune system in the gastrointestinal tract. Potassium chloride (KCl) is the most commonly used mineral salt due to its similarity to sodium chloride (NaCl), and its consumption has been shown to lower blood pressure when consumed in adequate amounts. The aim of this study was to partially replace NaCl with KCl at levels of 25%, 50% and 75% in the brine used to make white brined cheese. Parameters such as acidity, total dissolved solids, salinity, conductivity, colour, texture and sensory properties were evaluated during a 28-day refrigerated storage period. KCl can replace NaCl by 50%, and no significant differences in physicochemical and sensory parameters were observed during cheese storage compared to the control sample. In addition, the study investigates the use of Artificial Neural Network (ANN) models to predict certain brine and cheese properties. The study successfully developed four different ANN models to accurately predict various properties such as brine pH, cheese colour and hardness over a 28-day storage period.

## 1. Introduction

Numerous studies have shown that an increased intake of potassium in the diet can have a protective effect on people with sodium-induced high blood pressure. It reduces urinary calcium excretion, may contribute to the maintenance of skeletal mass and reduces the risk of kidney stone formation [1,2,3,4,5,6]. The World Health Organization (WHO) recommends a daily salt (NaCl) intake of less than 5 g for adults, which is equivalent to 2 g of sodium, but most people consume larger amounts of NaCl (9 to 15 g/day). To tackle this problem, WHO member states have introduced strategies aimed at reducing salt consumption by 30% by 2025 [7]. It is assumed that around 75% of sodium intake comes from processed foods [8].

With cheese consumption increasing worldwide, it is important to reduce the amount of salt used as a source of sodium without compromising the acceptability of the product. In cheeses such as cheddar, mozzarella and feta cheese, potassium chloride (KCl) can effectively reduce sodium content without major quality issues, although it can increase bitterness and interfere with biochemical reactions. Magnesium chloride (MgCl_2_) and calcium chloride (CaCl_2_) are less suitable due to off-flavours and lower acceptability of the cheese. Recent research suggests that calcium lactate could be a viable alternative, reducing the sodium content in Fourme d’Ambert by 19% and offering potential nutritional benefits [8]. In low-sodium dairy products, more than 25% of the NaCl is usually replaced by other salt alternatives. KCl, or a combination of NaCl and KCl, is commonly and successfully used as a partial substitute for NaCl in various cheeses, including feta cheese [6]. KCl is the most commonly used mineral salt due to its similarity to NaCl [9]. The USDA recommends increasing potassium intake for the general population, and higher dietary potassium intake has been shown to lower blood pressure when consumed in adequate amounts [10]. However, the introduction of potassium-fortified salts depends on factors such as their impact on taste, preservation, microstructure and cost. In particular, replacing NaCl with KCl may result in an undesirable bitter and metallic taste, which is recognised and not preferred by consumers. Katsiari et al. [11] demonstrated that it is possible to reduce the sodium content in feta cheese by up to 50% by partially replacing NaCl with KCl without any negative effect on quality. Their results showed that cheeses produced with NaCl/KCl mixtures showed no significant differences (*p* > 0.05) in terms of composition (moisture, fat, protein, salt), physicochemical properties (pH, a_w_), sensory attributes (appearance, body and texture, flavour, overall quality) and textural characteristics (breaking strength and pressure, hardness) compared to control cheeses. Golin Bueno Costa et al. [12] replaced NaCl with KCl (25% and 50%) in Minas Padrão cheese, where cheese became harder and firmer to the bite, and the fatty acid profile improved with more short-chain and polyunsaturated fats. A 25% substitution showed no significant taste differences compared to the control. Ferroukhi et al. [8] investigated different methods of salt reduction in blue cheese with regard to their effects on various properties. They found that reducing the salt content (from 11% to 67%) increased water activity but had no significant effect on the microbiota or hygienic quality of the cheese, apart from a decrease in yeasts and moulds.

Considering the complexity of cheese production with substitute salts, such as the variability in raw materials, pH values of brine, ripening time the interactions between curd and salt, alternative modelling approaches such as Artificial Neural Network (ANN) can be employed to predict some properties. ANN can handle complex, non-linear relationships within the data and is, therefore, suitable for modelling complicated processes in cheese production. In the context of cheese production with substitute salts, ANN can be used for both steady-state and dynamic modelling as well as for process optimisation. For example, a static feedforward neural network could be developed to model the effects of different substitute salts on the ripening process of cheese, including flavour development, texture and microbial activity. In addition, a moving window ANN could be used for real-time monitoring and dynamic modelling of critical cheese production variables such as pH, moisture content and salt diffusion [13]. By analysing sensor data, the ANN could estimate unmeasurable state variables and provide insights into how the substitute salts affect the physical and biochemical properties of the cheese throughout the production process.

Considering the importance of producing foods with lower sodium content and the limited research on low sodium cheese in brine, this study aimed to investigate whether NaCl can be partially replaced by KCl in the brine at high KCl concentration (up to 75%), which has not yet been demonstrated. Also, the study will evaluate whether ANNs are capable of capturing complex, non-linear relationships within the data and highlight their value in predicting brine pH, cheese colour and hardness over a 28-day storage period. The study focused on evaluating the effects of this substitution on the physicochemical and sensory properties of white brined cheese. Acidity, brine conductivity, salt quantity, texture, colour, protein content and sensory profile were determined. The kinetic models will be used to show whether the changes in brine properties such as pH value, TDS value, conductivity and colour parameters during the storage period could follow systematic and predictable trends.

## 2. Materials and Methods

### 2.1. Materials

The fresh cow’s milk for the production of white brined cheese was sourced from a nearby dairy farm. The mesophilic starter culture CHN-22 (IFF Danisco, Rheinmünster, Germany) was utilised in the cheese production process. Rennet (enzyme preparation BioRen^®^Classic 80LHA150, Chr. Hansen, Hoersholm, Denmark) was added according to the manufacturer’s instructions. The additives used included CaCl_2_ and KNO_3_, 9% acetic acid for lowering the milk’s pH, and the salts NaCl and KCl, both analytical grade.

### 2.2. White Brined Cheese Production

Each batch of cheese was made from 12 litres of raw milk. After sampling, the milk was standardised to about 3% milk fat by skimming in a separator (EP-80, Techtnika, Podplat, Slovenia) and pasteurised at 72 °C for 15 s in the water bath. The rest of the procedure was as described in our previous work by Lisak Jakopović et al. [14]. The cheese was ripened in brine and kept refrigerated for 28 days. Analyses were performed at 7-day intervals (before brining (day 0) and on days 7, 14, 21 and 28).

### 2.3. Brine Production

All brine solutions were prepared with a 10% salt concentration (*w*/*v*), and the pH was adjusted to 4.7 with 9% acetic acid. The control brine contained 10% NaCl (*w*/*v*) and was designated as sample BC. KCl was used to replace NaCl in the brine in a ratio of 25%, 50% and 75% (*w*/*v*), corresponding to sample names BK1, BK2 and BK3, respectively. After preparation, the brine was pasteurised at 72 °C for 15 s and then cooled to storage temperature. The ratio of brine to cheese was maintained at 4:1 during cheese immersion. The control cheese sample was labelled C, while the cheeses salted with 25%, 50% and 75% salt substitutes were labelled K1, K2 and K3, respectively.

### 2.4. Brine and Cheese Analyses

The white brined cheese was produced in three separate batches, with an interval of 14 days between each batch. Both the cheese and the brine were analysed on the day of production (day 0) and then every 7 days during the 28-day cold storage period. The acidity (pH and titratable acidity in Soxhlet-Henkel degrees, °SH) was determined according to the method described by Lisak Jakopović et al. [14]. Conductivity and total dissolved solids (TDS) were measured using a conductivity meter (TDS/Conductivity/°C meter, RS 232 CON 200 series, Oakton, Singapore) as described by Lisak Jakopović et al. [15]. The salt content was determined using the Mohr method (ISO 1738, 2004 [16]), in which chlorides were titrated with a silver nitrate solution in the presence of chromate anions and the end point of the titration was characterised by the appearance of red silver chromate [17].

### 2.5. Cheese Texture Analyses

Texture profile analysis (TPA) of the white brined cheese samples produced was performed using a texture analyser (Ametek Lloyd Instruments Ltd., West Sussex, UK). The analyser was equipped with a 50 kg load cell and operated with NEXYGEN Plus 3.0 software. The tests were performed at room temperature. Prior to analysis, samples were cut into 10 × 10 × 10 mm cubes and conditioned at 20 °C for 2 h. The samples were compressed twice at a crosshead speed of 1 mm/s until 50% deformation was reached, with a 5-s pause between cycles. The following parameters were calculated from the force-displacement curves: Hardness (N), Adhesive force (N), Adhesiveness (N mm), Cohesiveness (N/m), Gumminess (N), Postponed elasticity (mm), Chewiness (N mm), Resistance (N), Breakage (N) and Fibrousness (mm).

### 2.6. Colour Measurements

The CIE surface colour parameters (L*: lightness, a*: redness, b*: yellowness) (CIE, 1976) were measured using a Minolta CM-700d spectrophotometer (Konica Minolta, Tokyo, Japan) equipped with illuminant D65, a 10° standard observer, an 8 mm aperture and an open cone. Each cheese and brine sample was analysed in triplicate [18].

### 2.7. Sensory Analyses

The sensory evaluation of the white brined cheese was carried out by a panel of five specially trained tasters using a weighted scoring system on a 20-point scale [19,20]. The cheese was stored at 4 °C from the time of production, including the brine, until sampling and evaluation. In a room set up according to ISO standard 8589:2007 [21], the cheese samples were taken from the brine, coded, evenly portioned and presented to the panellists. Each sample was assessed for appearance, colour, consistency, cut, odour and taste, with attributes rated on a scale of 1 to 5. The average score for each attribute was multiplied by a predetermined weighting factor, resulting in the following scores for each attribute: Appearance—2, Colour—1, Consistency—2, Cut—3, Odour—2, and Taste—10. The final score for each sample was the sum of the scores for all attributes, with the maximum possible score being 20 [14].

### 2.8. Statistical Analysis and Data Modelling

#### 2.8.1. Basic Statistical Analysis and Analysis of the Variance

The preparation of white brined cheese was repeated three times, and the results of all measurements were expressed as mean ± standard deviation (SD). Basic statistical analysis, including means and standard deviations, was performed using the software Statistica 14.0 (Tibco Software Inc., Palo Alto, CA, USA). The means of the results were assessed using analysis of variance (ANOVA), and Tukey’s test was used to compare significant differences (*p* < 0.05) between the physicochemical, textural and sensory evaluations. ANOVA was performed using the software Statistica 14.0 (Tibco Software Inc., Palo Alto, CA, USA).

#### 2.8.2. Kinetics of Brine Properties Change over the Time

Kinetics of brine properties (pH, TDS, S, colour parameters (L*, a*, b*)) change over time was decried using the following models: (i) exponential growth model (Equation (1)), (ii) exponential growth decay (Equation (2)), (iii) exponential growth to maximum (Equation (3)) and (iv) quadratic model (Equation (4)):(1)$$Y=a\cdot e^{b\cdot t}$$
(2)$$Y=a\cdot e^{-b\cdot t}$$
(3)$$Y=a\cdot \left(1-e^{b\cdot t}\right)$$
(4)$$Y=a\cdot t^{2}+b\cdot t+c$$
where *Y* presents selected brine property (pH, TDS, S, colour parameters (L*, a*, b*)). Parameters of the models a, b, and c were estimated using non-linear regression implemented into software Statistica 14.0 (Tibco Software Inc., Palo Alto, CA, USA). The model performance was estimated using the coefficient of determination (R^2^), using adjusted coefficient of determination (R^2^_adj_) and using standard error of estimation (SEE).

#### 2.8.3. Principal Component Analysis (PCA)

Principal component analysis was performed to determine the effects of the process variables on all brine and cheese properties analysed. For the brine properties, NaCl concentration, KCl concentration and day of storage were used as additional variables, while pH, conductivity, TDS and colour coordinates (L*, a*, b*) served as variables for the analysis. Similarly, for cheese properties, NaCl concentration, KCl concentration, day of storage, brine pH, brine conductivity and brine TDS were used as supplementary variables, while cheese pH, °SH, cheese colour coordinates (L*, a*, b*), the textural properties of the cheese (hardness, adhesive force, adhesiveness, cohesiveness, gumminess, postponed elasticity, chewiness, resistance, breakage and fibrousness) and the sensory properties of the cheese (appearance, colour, consistency, cut, smell, taste and overall impression) were the variables to be analysed.

#### 2.8.4. Artificial Neural Network (ANN) Modelling

ANN modelling was employed to predict the properties of brine, as well as the physical properties of cheese, total colour change of cheese and cheese hardness over the 28-day storage period. Multiple Layer Perceptron (MLP) networks were developed using the software Statistica 14.0 (TIBCO^®^ Statistica, Palo Alto, CA, USA). ANNs consist of three layers: input, hidden, and output. Four different ANN models were developed:(i)For prediction of brine properties (pH, conductivity, TDS and colour coordinates) based on NaCl concertation, KCl concentration and day of storage. The dataset for the construction of ANNs was 60 × 9, with 60 rows representing brine samples, 3 columns representing model inputs, and 6 columns representing model output;(ii)For prediction of physical properties of cheese (pH, °SH, L and colour coordinates) based on NaCl concertation, KCl concentration, day of storage, brine pH, brine conductivity and brine TDS. The dataset for the construction of ANNs was 60 × 9, with 60 rows representing brine samples, 6 columns representing model inputs, and 3 columns representing model outputs;(iii)For prediction of cheese total colour change based on NaCl concertation, KCl concentration and day of storage, brine pH, brine conductivity and brine TDS. The dataset for the construction of ANNs was 60 × 7, with 60 rows representing brine samples, 6 columns representing model inputs, and 1 column representing model output;(iv)For the prediction of cheese hardness based on NaCl concertation, KCl concentration, day of storage, brine pH, brine conductivity and brine TDS, the dataset for the construction of ANNs was 60 × 7, with 60 rows representing brine samples, 6 columns representing model inputs, and 1 column representing model output.

To ensure the development of the relabel models, cross validation implemented in Statistica 14.0 (TIBCO Statistica, Palo Alto, CA, USA) was used. A subsampling strategy was applied. The back error propagation algorithm was used for model training, and the error function was the sum of squares. R^2^ and Root Mean Square Error (RMSE) values for training, testing, and validation were used to estimate the performance of developed models. In order to develop ANN, data were randomly divided into three categories: network training (70%—42 data points), model test (15%—9 data points), and model validation (15%—9 data points).

## 3. Results and Discussion

### 3.1. Physicochemical Properties of Brine

Table 1 shows the physicochemical properties of the control brine (BC), and the brines prepared with KCl (BK1, BK2, BK3) during a 28-day cold storage period. The pH value in the BC remained relatively stable until the seventh day of storage but then increased significantly, reaching 4.99 at the end of the storage period. In contrast, the pH value in the brines with KCl substitution (BK1, BK2, BK3) increased significantly on the seventh day of storage, and this increasing trend continued in all brines until the end of the storage period. The highest pH value of 5.23 pH units was measured in brine BK3 (KCl, 75%). When looking at the significance between the brines on the same day of storage, it can be seen that there is no difference between BC and BK1. In contrast, a significant difference was found between BC and BK2 and BK3.

TDS represents the total concentration of dissolved solids in the analysed liquid. All brine types were prepared with the same amount of salt (10% *w*/*v*), and no significant difference in TDS values was found between the different brine types. In our previous study [14], when calcium citrate and calcium lactate were used as replacement salts, a significant difference was found between the different brines because calcium salts do not dissolve as easily as KCl. In addition, the brine did not remain trans-parent when calcium salts were used, as was the case when KCl was used. The brines prior to cheese immersion in BK1 and BK2 had the highest TDS values, 55.33 g/L and 55.63 g/L, respectively. For all types of brine, the TDS values decreased significantly during the 28 days of cold storage, reaching their lowest value on the 28th day. The decrease in TDS can be attributed to the migration of salt into the cheese.

In general, the conductivity depends on the NaCl concentration in the solution; higher NaCl concentrations are directly proportional to the conductivity values [22]. The conductivity of the salt solution followed the same trend as the TDS values for all sample types. The highest conductivity was observed for all samples before the cheese was immersed in the brine (between 96.60 and 111.23 mS/cm), and no significant differences were observed when comparing the brines for each storage day. In addition, there was a significant decrease in conductivity during the storage period and cheese ripening for both the control and other samples. This decrease can be attributed to the migration of NaCl from the brine into the cheese, as the electrical conductivity is directly related to the ion concentration in the solution. During the storage period, conductivity decreased significantly in both the control and other brine samples, with the highest conductivity values observed in brines BK1 and BK2. In our previous studies [14], the conductivity values of brines with substitution salts (calcium citrate and calcium lactate) were significantly lower than those of control brines during the storage period when different substitution salts were used. The lowest conductivity of 59.10 mS/cm was observed on the 28th day of cold storage in the brine in which 75% of the NaCl was replaced by KCl. In contrast, the lowest conductivity value when calcium salt was substituted was 37.10 mS/cm when 50% of the NaCl was replaced with calcium lactate [14]. In general, KCl contains potassium (K^+^) and chloride (Cl^−^) ions, while calcium salts such as calcium citrate or calcium lactate contain calcium (Ca^2+^) ions. Potassium ions are smaller and more mobile in water compared to calcium ions, which leads to a higher electrical conductivity of KCl compared to calcium salts [23].

Based on the kinetic models for the changes in brine properties shown in Table 2, some observations can be made about the behaviour of various parameters during the storage period. For all samples, the pH showed exponential growth over time. The growth rate constants (b) were relatively consistent, with values around 0.0021–0.0029 1/day for all samples. This indicates a gradual increase in pH, which could indicate a decrease in acidity during the ripening process. The high R^2^ values (above 0.8 for most samples) mean that the model fits the data well, indicating consistent pH changes over time. TDS showed an exponential decay with decay rate constants (b) between 0.0137 and 0.0157 1/day. This trend indicates a decrease in solute concentration, probably due to the diffusion of salts into the cheese matrix [24]. The R^2^ values of the model were remarkably high (above 0.85), suggesting that the decrease in TDS is a predictable and consistent process across different samples. Like TDS, conductivity also decreased exponentially, with rates of decrease between 0.0139 and 0.0177 1/day. The decrease in conductivity can be attributed to the reduction in ion concentration in the brine when salts migrate into the cheese during storage time. Again, the R^2^ values showed good agreement, confirming that the exponential decay model effectively describes the conductivity changes during storage. Colour parameter L* showed an exponential decay across all samples, indicating a gradual darkening or reduction in lightness over time. This could be due to changes in the cheese surface or interactions between the cheese and the brine [25]. Red–green component a* showed exponential growth, indicating an increase in red hues during storage. This change could be related to the development of certain pigments or microbial activity [26]. Yellow–blue component b* followed either quadratic models or exponential growth to a maximum, indicating complex changes in yellow–blue tones. This variability could reflect different compositional changes or the impact of different brine compositions on the evolution of cheese colour. Overall, the kinetic properties of the brine show that most changes are systematic and predictable and are influenced by the interactions between the brine and the cheese over time. The use of kinetic models helps to clarify the rate and nature of these changes and supports further optimisation of brine processes to achieve the desired cheese characteristics.

### 3.2. Physicochemical Properties of Cheese

The results in Table 3 show the effects of storage time on various physicochemical parameters for the control sample (C) and the samples in which NaCl was replaced by KCl at levels of 25% (K1), 50% (K2) and 75% (K3) over a period of 28 days. Each parameter (pH, °SH (acidity), salt concentration and colour (L*, a*, b*)) was measured before brining and after 7, 14, 21 and 28 days of cold storage in brine. The results provide information on the stability and quality changes during storage, although there are clear differences between the samples.

The pH values of all samples showed a general increasing trend over the 28-day period, which is consistent with typical storage-related shifts due to microbial activity and chemical reactions. On day 0, the pH of samples C, K1 and K2 was identical (4.85 ± 0.10), while K3 had a slightly higher pH of 5.04 ± 0.00. Over time, the pH of all samples increased, with sample C reaching 5.25 ± 0.16 on day 28, while K3 had the highest final pH of 5.39 ± 0.00. The steady increase in pH, especially in sample K3, could indicate greater microbial or enzymatic activity, possibly due to higher buffering capacity or other microbial dynamics. Also, in sample K3, 75% of the NaCl was replaced by KCl and the initial pH of the sample was higher compared to the other samples. Potassium ions (K⁺) have different properties compared to sodium ions (Na⁺). KCl inhibits the activity of certain bacteria that produce alkaline substances less effectively than NaCl. This bacterial activity can lead to a higher pH in cheese when a significant proportion of NaCl is replaced by KCl, as in our study (75%) [27]. In addition, the substitution of NaCl by KCl can alter the proteolysis process in cheese. KCl may affect enzyme activity and microbial metabolism differently than NaCl, leading to changes in the breakdown of proteins into peptides and amino acids, some of which may contribute to an increase in pH [28]. The °SH values, which represent titratable acidity, showed a significant decrease in all samples during storage. At the beginning, samples C, K1 and K2 had identical °SH values (83.20 ± 0.00), while K3 started with a significantly (*p* > 0.05) lower value (63.20 ± 0.00). This initial difference suggests that sample K3 may have had a lower initial acidity or a different acidification process due to the highest amount of KCl. Over time, the acidity of all samples decreased significantly (*p* > 0.05), with sample C decreasing to 35.90 ± 1.40 and K3 to 26.40 ± 0.40 by day 28. The faster decrease of °SH in K3 could correlate with the initial lower acidity, which could affect the preservation properties.

On day 7, sample K1 with 25% KCl and 75% NaCl had the highest salt concentration (6.37 ± 0.06), followed by K2 with 50% KCl (6.24 ± 0.08). Sample C, which contained 100% NaCl, had a lower salt concentration (5.46 ± 0.26), while K3, which had the highest KCl content at 75%, had the lowest salt concentration (5.11 ± 0.27). This pattern continued with slight variations during the storage period, with the samples with higher KCl substitution generally having lower or more stable salt concentrations than the 100% NaCl sample. The trend towards lower initial salt concentrations in samples with partial KCl substitution (K1 and K2) compared to the 100% NaCl sample (C) can be explained by the different physicochemical behaviour of NaCl and KCl. KCl has a higher molecular weight than NaCl, which results in lower molar concentrations and, thus, different ionic strengths in the cheese matrix when NaCl is substituted by KCl [27]. In addition, KCl could change the water binding capacity of the cheese, which affects the apparent salt concentration in the aqueous phase. Sample K3, with the highest KCl content (75%), had the lowest salt concentration determined by the Mohr method. This could be due to the fact that KCl dissolves less well in the cheese matrix compared to NaCl, possibly due to differences in solubility or interaction with the protein network of the cheese. Studies have shown that KCl may interact differently with casein micelles, which could affect the overall perception and measurement of salinity [27]. Over the 28 days, the salt concentration in sample C (100% NaCl) increased more significantly (*p* > 0.05) than in the other samples, reaching 6.83 ± 0.51 on day 28, suggesting that NaCl may diffuse more easily into the cheese matrix during storage and stabilise there. In contrast, the salt concentration in K1, K2 and K3 fluctuated less, with K3 having a relatively stable but lower salt concentration throughout the storage period.

Colour is an important quality feature that influences consumer perception. Lightness values (L*) varied slightly between samples, with K1 and K3 generally having higher L* values, indicating less browning or colour deterioration [29]. On day 28, sample K1 had a lightness value of 90.61 ± 0.59, while K2 had a slightly higher L* value of 93.27 ± 0.30, suggesting better colour stability. The a* and b* values, which represent the red–green and yellow-blue components, respectively, showed no significant changes (*p* > 0.05), indicating that the colour stability of the samples was relatively well-maintained during storage.

### 3.3. Textural Properties of Cheese

Texture is a main quality attribute of cheese and is defined as a composite sensory characteristic resulting from the tactile, visual and oral manifestation during consumption. It is influenced by some factors such as salt, protein, fat content, pH and mineral composition [30].

Table 4 shows the effects of replacing NaCl with KCl on various texture parameters of the cheese during a 28-day storage period. A control sample (C) with 100% NaCl and three test samples in which NaCl was replaced by KCl at 25% (K1), 50% (K2) and 75% (K3) were compared. Texture parameters such as hardness, adhesive force, adhesiveness, cohesiveness, gumminess, chewiness, resistance, breakage, and fibrousness were measured after 7, 14, 21 and 28 days. The hardness of the samples decreased over the storage period for all samples. On day 7, K1 had the highest hardness (14.46 ± 4.95 N), while K3 had the lowest hardness (8.42 ± 1.27 N). This indicates that a higher KCl concentration could lead to a softer texture. As NaCl reduces the water activity, the cheese often has a firmer and more compact texture due to the stronger interaction between casein proteins and the reduced water content. KCl has a different ionic effect on the cheese proteins and moisture content and contributes to a softer texture [31,32]. On day 28, all samples showed a significant decrease (*p* > 0.05) in hardness, with sample C having a higher hardness (5.21 ± 0.80 N) than K1 (4.27 ± 0.57 N). The decrease in hardness could be due to progressive proteolysis and moisture migration, processes known to soften cheese over time. The adhesive force, i.e., the force required to overcome the adhesive interaction between the cheese and the test probe, was relatively constant for all samples and throughout the storage period. Adhesiveness varied slightly but showed no clear trend depending on the amount of KCl used. Sample K3 exhibited the highest adhesiveness on day 7 (0.77 ± 0.25 N mm) and maintained a relatively high adhesiveness throughout the study. This could indicate that a higher KCl content affects the ability of the cheese matrix to bind water, resulting in increased adhesiveness. Since the cheese matrix retains more moisture in the presence of KCl, this additional water can lead to increased softness and stickiness (adhesiveness) of the texture. This is due to the fact that the looser protein structure means that more water remains trapped in the matrix [32]. Cohesiveness remained stable for all samples during storage, with values ranging from 0.21 ± 0.02 N/m to 0.28 ± 0.05 N/m. There were no significant differences between the samples, indicating that the replacement of NaCl with KCl did not significantly affect the structural integrity of the cheese. Gumminess decreased with time in all samples, particularly in K1 and K3, which had the lowest values on day 28 (0.96 ± 0.16 N and 0.92 ± 0.12 N, respectively). Chewiness followed a similar pattern to gumminess, with a marked decrease over time. On day 28, the chewability was significantly (*p* > 0.05) reduced for all samples, with K3 showing the lowest value (2.09 ± 0.83 N mm). The resistance, i.e., the force required to deform the cheese, was relatively the same for all samples. Although the resistance decreased slightly over time, there was no significant difference (*p* > 0.05) between the control and KCl-substituted samples. This indicates that the resistance is not significantly affected by the replacement of NaCl with KCl. Breakage, which indicates the force at which the cheese structure fails, decreased significantly (*p* > 0.05) over the storage period. On day 28, the control sample (C) again exhibited slightly higher breakage values (4.58 ± 0.57 N) than the K3 sample (3.23 ± 0.65 N). This reduction in breakage strength indicates that a higher substitution of KCl may lead to a more brittle cheese structure over time. The fibrousness, i.e., the extent to which the cheese stretches or forms fibres, varied but remained relatively stable across all samples. The highest fibrousness was observed in K3 on day 7 (10.22 ± 0.43 mm) and day 14 (11.74 ± 1.30 mm). These results suggest that KCl promotes a more fibrous texture, which could be related to its effect on the protein matrix and water binding in cheese.

Similar textural properties were observed in our previous work [14] when NaCl was partially replaced by calcium salts (calcium lactate and calcium citrate). It was concluded that during the ripening period, casein hydrolysis or dissolution of some of the residual colloidal calcium phosphate in the casein matrix of the cheese could have contributed to a decrease in hardness values of all cheese samples.

### 3.4. Sensory Properties

Table 5 shows the results of the sensory evaluation of the cheese samples, including appearance, colour, consistency, cut, odour, and taste after the 7th, 14th, 21st and 28th days. The values for all samples in terms of appearance remain relatively constant during the 28 days and show no significant differences (*p* >0.05). This may be due to the presence of a greater number of cracks on the surface of the cheese and the deviation from the snow-white colour of the cheese, as some samples were rather yellowish. Sample C, for example, shows only a minimal change and maintains a value of 1.9 ± 0.1 on day 7 and 1.9 ± 0.2 on day 28. Samples K1, K2 and K3 also show slight fluctuations, with K2 showing a decrease (1.7 ± 0.5) on day 21, indicating some fluctuation.

The colour measurements show a slight decrease over time, especially in samples K2 and K3. Initially, all samples had similar colour scores, but by day 28, K2 and K3 recorded values of 0.8 ± 0.2 and 0.9 ± 0.1, respectively. The colour change can be influenced by the colour of the brine in which the cheese was stored.

No significant difference (*p* > 0.05) was found in the consistency of cheese samples K1, K2 and K3 compared to the control sample (C). These cheeses were described as softer by the sensory analysts compared to the control sample. Cheese K3 received the lowest consistency rating (1.7 ± 0.6) on the 28th day of storage, as it was too soft in the opinion of the testers. With increasing storage time, the hardness of the cheese decreases, which is due to a higher salt content in the cheese mass, as confirmed by the texture results shown in Table 4.

The cut of the cheese was satisfactory in all samples and did not differ significantly from the control sample. The smell of all cheese samples was characterised as typical for white brined cheese and did not differ significantly (*p* > 0.05) from the control sample. The odour of cheese samples showed only slight fluctuations during the entire storage period. Sample C consistently shows a value of about 2.0, while K1, K2 and K3 also remain stable, which indicates that the odour of the samples is not significantly (*p* > 0.05) influenced by the storage time.

Taste is the most important sensory characteristic of white brined cheese and has the greatest influence on the overall evaluation of the sample. White brined cheese is characterised by a salty, sour and slightly pungent taste, which is enhanced by the ripening process. The taste scores show greater variability and significant differences (*p* > 0.05), especially in sample K3, which is 7.8 ± 2.7 on day 7 and decreases sharply to 4.6 ± 2.8 by day 28, which was classified as too salty and very bitter. In contrast, samples C, K1 and K2 retained their higher taste values. It can, therefore, be concluded that an excessive amount of KCl in the brine leads to an undesirable and bitter taste, which is consistent with the studies on fresh cheese [33].

### 3.5. Principal Component Analysis

PCA analysis presented in Figure 1 offers a detailed overview of the relationships between various brine and cheese properties throughout the ripening process. According to Silva Matinis et al. [34], PCA is a method used to identify the key factors driving changes in industrial food products. In this study, PCA was conducted to assess the impact of NaCl and KCl on these properties, with the goal of understanding how these factors contribute to the overall variance in the dataset. The plot illustrates two main components: PC 1, which accounts for 77.65% of the variance, and PC 2, which explains a further 13.36% of the variance. Together, these components provide a clear representation of the primary and secondary variations within the dataset and contribute to 91.01% of the data variability. The high percentage of variance captured by PC 1 indicates that it is the dominant source of variability and reflects the most important trends among the measured traits. Several key variables are plotted along these axes. TDS and salt concentration (S) exhibit a strong positive correlation with PC 1, suggesting that changes in TDS and salt concentration are the main drivers of the observed variations in brine properties. Lightness (L*), which also shows a positive correlation with PC 1, suggests that changes in these properties are closely linked to cheese ripening. Conversely, pH, the red–green colour component (a*), and the yellow–blue component (b*) are positioned on the left side of the diagram, indicating a negative correlation with PC 1. These variables show a more complex relationship and contribute to variance in a direction opposite to that of TDS and S. This suggests that while salinity and lightness are associated with the primary variation, pH and colour changes move in the opposite direction. The points labelled KCl and NaCl represent samples or conditions associated with potassium chloride and sodium chloride, respectively. The positioning of KCl closer to the pH and colour variables suggests that KCl may have a stronger influence on these properties than NaCl. In contrast, NaCl is closely related to TDS and salt concentration, highlighting its role in controlling these specific properties during cheese ripening. Additionally, the variable time (t) is linked to pH and colour components, indicating that changes over time significantly affect these properties during storage. Overall, the PCA plot underscores the significant impact of salt type and concentration on the physicochemical properties of the cheese and highlights the complex interplay of these factors in determining the final product quality.

Figure 2 presents PCA relationships between the sensory and physico-chemical properties of cheese when NaCl is partially replaced by KCl. These analyses aim to clarify the effects of salt substitution on cheese quality and the influence of various properties. Both plots identify two principal components that capture the majority of the variance within the dataset and provide insights into the underlying factors that drive changes in cheese properties. The first principal component (PC 1) accounts for a significant proportion of the variance (28.93% in both figures) and reflects the main source of variation across the parameters assessed. PC 2 captures a further 13.13% of the variance in the first plot and 13.87% in the second plot, indicating secondary variation. Plots contribute to 42.23% and 42.80% of the data variability, respectively. In the first PCA plot, PC 1 primarily differentiates properties related to salt concentration, appearance and adhesive force, with these variables positively aligned along this axis. This indicates that a higher NaCl content improves these properties, which are crucial for the overall acceptability and quality of the cheese. In particular, the position of NaCl in the plot confirms the positive association with these desirable properties and emphasises its role in maintaining cheese quality. Conversely, KCl is on the negative side of PC 1 and correlates negatively with colour change (∆E) and odour. This positioning indicates that replacing NaCl with KCl may have a negative impact on the sensory perception of the cheese, possibly leading to less favourable colour and aroma profiles. These results highlight a trade-off between reducing sodium content and maintaining sensory quality. PC 2 in the first plot illustrates the impact of NaCl on texture-related properties such as resistance and cohesion. These properties are positively positioned along this axis, indicating that NaCl contributes to the improvement of texture quality. The inverse relationship with taste and consistency, which is negatively positioned along PC 2, suggests that these sensory attributes may be affected by salt substitution. The second PCA plot shows a similar pattern, with PC 1 capturing the variance associated with salt concentration and appearance, while PC 2 highlights the changes in textural properties. In this plot, the addition of brine-related variables such as brine salt concentration and total dissolved solids in brine provides further context to the effects of NaCl and KCl on cheese quality. The presence of salt in the brine and brine TDS in the second plot underlines their importance in the overall matrix of cheese properties, especially their positive correlation with PC 2, suggesting that these brine properties significantly influence the texture and cohesion of the cheese, emphasising the importance of brine composition for cheese ripening and quality. KCl remains negatively associated with colour change and odour, similar to the first plot. However, in this plot, the pH of the brine and the pH of the cheese are closely related to KCl, suggesting that its addition can alter the acid-base balance in the cheese matrix, potentially affecting the microbial activity and ripening processes of the cheese [35].

In conclusion, the PCA reveals significant insights into the effects of partially replacing NaCl with KCl on the sensory and physico-chemical properties of cheese. The findings suggest that while KCl can be used as a partial substitute for NaCl, careful consideration is needed to balance sodium reduction with the preservation of cheese quality, especially in terms of sensory attributes and texture.

### 3.6. Artificial Neural Network Modelling

ANN modelling was used to predict the properties of brine and the physical and textural properties of cheese over a 28-day storage period. ANNs are adept at capturing and modelling non-linear relationships within the data. This is particularly valuable in big data, where relationships between variables are often complex and not easily captured by traditional linear models [36]. Four different ANN models were developed for this study. The first model was used to predict brine properties such as pH, conductivity, TDS and colour coordinates based on variables such as NaCl concentration, KCl concentration and day of storage. The second model was used to predict the physical properties of the cheese, including pH, °SH, L, and colour coordinates, using inputs such as NaCl concentration, KCl concentration, day of storage, brine pH, and brine conductivity. The third model focused on predicting the cheese colour change based on NaCl concentration, KCl concentration, day of storage, brine pH and brine conductivity. Finally, the fourth model was developed to predict the cheese hardness using the same set of input variables. The architectures of the ANNs chosen to describe specific properties of brine and white brined cheese are detailed in Table 6. To predict the physical properties of brine, the MLP 3-9-6 network was selected, utilising the Tanh function as the hidden activation function and the Logistic function as the output activation function. The model’s learning, testing, and validation R^2^ values were 0.9722, 0.9398, and 0.9048, respectively, with corresponding RMSE values of 0.0107, 0.0184, and 0.0373. This selected ANN model accurately describes all analysed outputs. The strongest agreement between the experimental data and the ANN model predictions for brine physical properties was observed for pH (R^2^training = 0.9892, R^2^test = 0.9883, R^2^validation = 0.9782) (Figure 3a), followed by b* (R^2^training = 0.9785, R^2^test = 0.9650, R^2^validation = 0.9456) (Figure 3f), and a* (R^2^training = 0.9626, R^2^test = 0.9605, R^2^validation = 0.9030) (Figure 3e). Conversely, the lowest agreement between the experimental and ANN-predicted data was observed for L* (R^2^training = 0.9551, R^2^test = 0.8069, R^2^validation = 0.7655) (Figure 3d).

To predict the physical properties of cheese, the MLP 6-4-3 network was selected, utilising the Tanh function as the hidden activation function and the Identity function as the output activation function. The model’s learning, testing, and validation R^2^ values were 0.8779, 0.8779, and 0.8770, respectively, with corresponding RMSE values of 0.0295, 0.0299, and 0.0468. This selected ANN model demonstrates high precision in describing all analysed outputs. The strongest agreement between the experimental data and the ANN model predictions for cheese physical properties was observed for salt concentration (R^2^ training = 0.9012, R^2^ test = 0.8525, R^2^ validation = 0.8425) (Figure 4c). Conversely, the lowest agreement was found for °SH (R^2^ training = 0.8574, R^2^ test = 0.8539, R^2^ validation = 0.8049) (Figure 4b). Additionally, highly efficient ANN models were developed for predicting cheese total colour change (MLP 6-7-1, Figure 4d) and cheese hardness (MLP 6-8-1, Figure 4e). In conclusion, the selected ANN models demonstrate a high level of precision in predicting the physical properties of both brine and white brined cheese. Additionally, the development of efficient models for predicting total color change and cheese hardness further underscores the capability of ANNs in capturing intricate relationships within the dataset, making them valuable tools for analysing and predicting key characteristics in food science.

## 4. Conclusions

The study shows that replacing NaCl with KCl in cheese brining significantly affects pH, acidity, salt concentration and colour stability over 28 days, with higher KCl content resulting in higher pH, lower salt concentration and consistent colour stability due to differences in microbial activity, ion properties and interaction with the cheese matrix. The kinetic models show that the changes in brine properties such as pH, total dissolved solids, conductivity and colour are systematic and predictable and are influenced by the interactions between the brine and the cheese during storage, which underlines the effectiveness of the models in optimising brine processes to achieve the desired cheese properties. While a reduction in sodium content is desirable from a health perspective, the analyses highlight the potential challenges in maintaining optimal sensory and textural qualities. The partial substitution of NaCl by KCl in cheese production leads to noticeable changes in textural properties during storage. A higher KCl content tends to reduce hardness, gumminess, chewiness, and breakage strength while potentially increasing adhesiveness and fibrousness. The research results underline the importance of comprehensive assessments when introducing salt substitution strategies in cheese production. In our earlier work, the substitute salts Ca-lactate and Ca-citrate also proved to be excellent for use in the production of sodium-reduced white brined cheese. The results showed that the replacement of 25% NaCl with substitute salts had no effect on the physicochemical, textural, sensory and colour properties of the cheese, just as in this work with KCl replacement. The selected ANN models show high accuracy in predicting the physical properties of brine and white brined cheese. The results show that ANNs are capable of capturing complex, non-linear relationships within the data and highlight their value in food science for analysing and predicting important properties. Further research could focus on optimising the balance between sodium reduction and maintaining desired cheese qualities, possibly investigating alternative salts or complementary techniques to improve sensory acceptability while minimising the sodium content.

## Figures and Tables

**Figure 1 foods-13-03031-f001:**
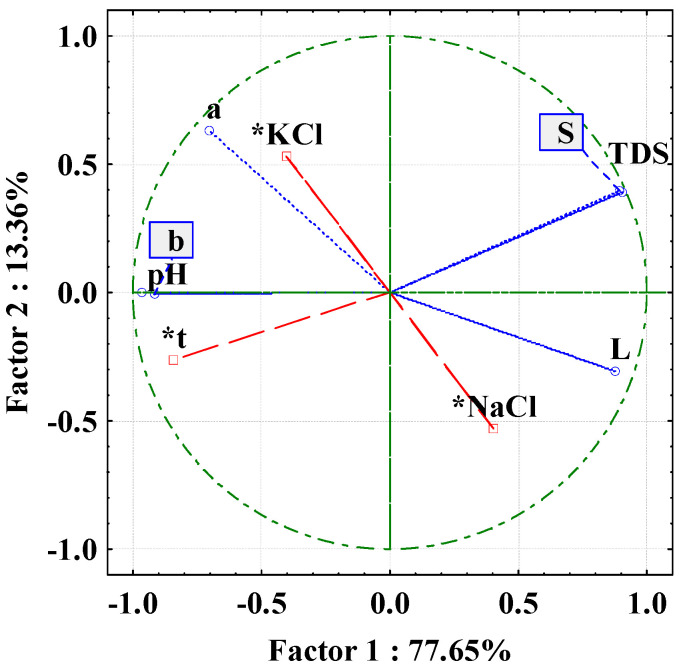
Principle component analysis (PCA) of brine properties during storage time (salt concentration (S), colour (L*, a*, b*)). * Supplementary variables.

**Figure 2 foods-13-03031-f002:**
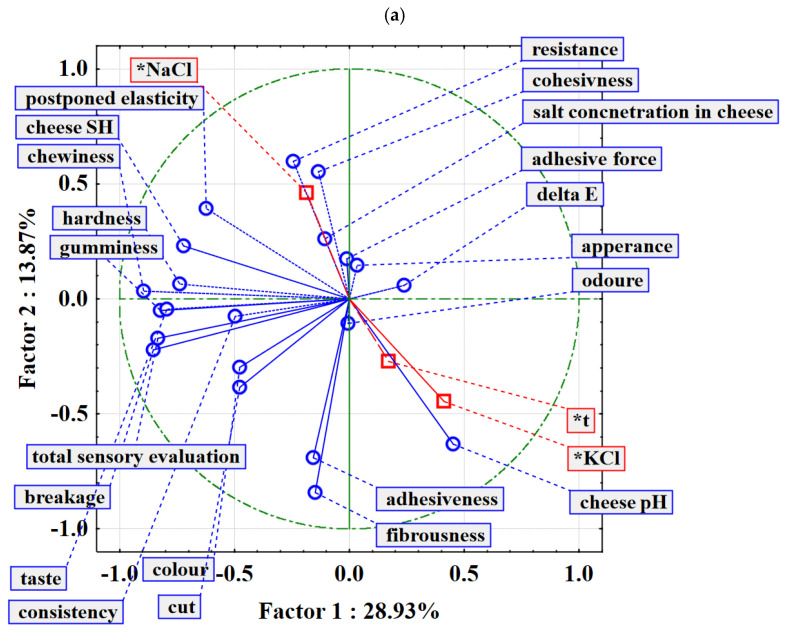
Principle component analysis (PCA) of cheese properties during storage time. (**a**) PC 1 differentiates properties related to salt concentration, appearance and adhesive force properties, PC 2 illustrates the impact of NaCl on texture-related properties such as resistance and cohesion; (**b**) PC 1 capturing the variance associated with salt concentration and appearance, PC 2 highlights the changes in textural properties. * Supplementary variables.

**Figure 3 foods-13-03031-f003:**
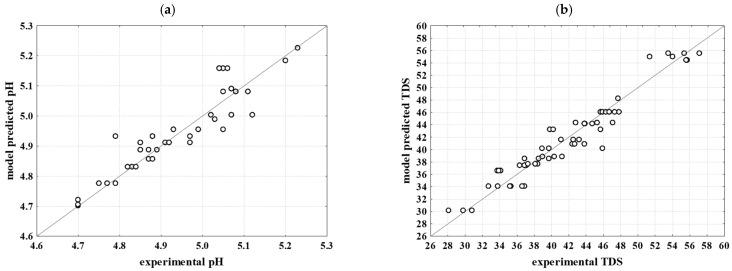
Comparison between experimental data and model-predicted data for brine properties. (**a**) pH value; (**b**) total dissolved solids; (**c**) salt concentration; (**d**) color—L*; (**e**) color—a*; (**f**) color—b*.

**Figure 4 foods-13-03031-f004:**
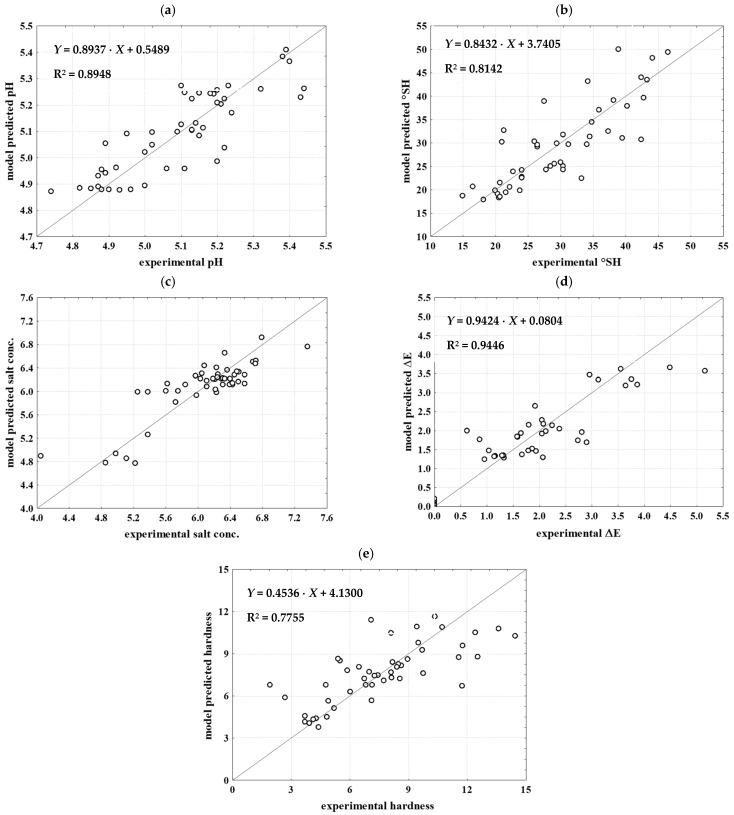
Comparison between experimental data and model-predicted data for cheese properties. (**a**) pH value; (**b**) titratable acidity (°SH); (**c**) salt concentration; (**d**) color change (ΔE); (**e**) texture—hardness.

**Table 1 foods-13-03031-t001:** Mean values for physical characteristics (pH value, total dissolved solids (TDS), conductivity (S), colour (L*, a*, b*)) of brine during white brined cheese storage at 4 °C for 28 days.

Variable	Day of Storage	Samples			
		BC	BK1	BK2	BK3
pH	0	4.70 ± 0.00 ^A,a^	4.70 ± 0.00 ^A,a^	4.70 ± 0.00 ^A,a^	4.70 ± 0.00 ^A,a^
	7	4.77 ± 0.02 ^A,a^	4.83 ± 0.01 ^A,b^	4.87 ± 0.03 ^B,b^	5.07 ± 0.05 ^C,b^
	14	4.88 ± 0.09 ^A,b^	4.87 ± 0.02 ^A,c^	4.88 ± 0.02 ^A,c^	5.08 ± 0.03 ^B,c^
	21	4.91 ± 0.06 ^A,c^	4.92 ± 0.00 ^A,d^	5.03 ± 0.07 ^B,d^	5.05 ± 0.01 ^C,d^
	28	4.99 ± 0.06 ^A,d^	5.07 ± 0.00 ^A,e^	5.20 ± 0.04 ^B,e^	5.23 ± 0.00 ^C,e^
TDS (g/L)	0	52.27 ± 2.89 ^A,a^	55.33 ± 1.81 ^A,a^	55.63 ± 0.06 ^A,a^	47.70 ± 0.00 ^A,a^
	7	44.13 ± 2.53 ^A,b^	45.97 ± 0.31 ^A,b^	47.27 ± 0.55 ^A,b^	40.17 ± 1.11 ^A,b^
	14	42.27 ± 2.48 ^A,c^	42.97 ± 0.74 ^A,c^	45.07 ± 2.15 ^A,c^	33.90 ± 0.20 ^B,c^
	21	38.37 ± 5.87 ^A,d^	41.50 ± 3.83 ^A,d^	41.90 ± 3.29 ^A,d^	36.23 ± 0.91 ^A,d^
	28	33.93 ± 3.35 ^A,e^	36.77 ± 0.42 ^A,e^	37.93 ± 0.56 ^A,e^	29.57 ± 1.37 ^A,e^
S (mS/cm)	0	103.97 ± 2.89 ^A,a^	110.83 ± 3.50 ^A,a^	111.23 ± 0.12 ^A,a^	96.60 ± 0.00 ^A,a^
	7	86.57 ± 2.57 ^A,b^	91.07 ± 0.61 ^A,b^	93.83 ± 0.61 ^A,b^	80.13 ± 1.33 ^A,b^
	14	83.93 ± 2.37 ^A,c^	87.37 ± 2.28 ^A,c^	90.70 ± 3.75 ^A,c^	68.37 ± 0.23 ^B,c^
	21	78.07 ± 4.10 ^A,d^	83.77 ± 7.09 ^A,d^	85.63 ± 6.51 ^A,d^	66.50 ± 1.40 ^B,d^
	28	67.75 ± 3.85 ^A,e^	74.07 ± 0.72 ^A,e^	76.70 ± 0.87 ^A,e^	59.10 ± 1.85 ^A,e^
L*	0	99.84 ± 0.11 ^A,a^	99.71 ± 0.00 ^A,a^	99.67 ± 0.22 ^A,a^	99.45 ± 0.00 ^A,a^
	7	96.77 ± 0.13 ^A,a^	92.99 ± 5.11 ^A,b^	93.33 ± 5.47 ^A,a^	82.67 ± 5.95 ^A,a^
	14	91.94 ± 1.77 ^A,a^	78.94 ± 10.37 ^A,a^	87.66 ± 1.20 ^A,a^	75.19 ± 17.71 ^A,b^
	21	92.64 ± 0.42 ^A,a^	81.26 ± 5.35 ^A,a^	78.47 ± 1.87 ^A,b^	77.73 ± 11.11 ^A,c^
	28	92.36 ± 0.60 ^A,a^	82.13 ± 8.51 ^A,a^	48.62 ± 3.26 ^B,c^	51.70 ± 7.85 ^C,d^
a*	0	0.03 ± 0.00 ^A,a^	0.05 ± 0.04 ^A,a^	0.08 ± 0.02 ^A,a^	0.07 ± 0.00 ^A,a^
	7	−0.31 ± 0.04 ^A,b^	0.17 ± 0.09 ^B,a^	0.16 ± 0.04 ^C,a^	0.33 ± 0.18 ^D,b^
	14	−0.29 ± 0.08 ^A,c^	0.32 ± 0.07 ^B,b^	0.20 ± 0.05 ^C,a^	0.54 ± 0.15 ^D,c^
	21	−0.24 ± 0.07 ^A,d^	0.26 ± 0.07 ^B,a^	0.31 ± 0.11 ^C,a^	0.51 ± 0.18 ^D,d^
	28	−0.12 ± 0.03 ^A,a^	0.36 ± 0.02 ^B,c^	0.55 ± 0.03 ^C,b^	0.82 ± 0.09 ^D,e^
b*	0	0.08 ± 0.02 ^A,a^	0.17 ± 0.08 ^A,a^	0.23 ± 0.04 ^A,a^	0.45 ± 0.00 ^A,a^
	7	3.77 ± 0.14 ^A,b^	4.38 ± 0.69 ^A,b^	4.56 ± 1.16 ^A,b^	6.07 ± 0.64 ^A,b^
	14	4.93 ± 0.34 ^A,c^	6.96 ± 0.42 ^A,c^	6.85 ± 0.62 ^A,c^	6.19 ± 1.58 ^A,c^
	21	5.96 ± 0.59 ^A,d^	7.07 ± 0.57 ^A,d^	8.02 ± 1.05 ^A,d^	7.43 ± 0.97 ^A,d^
	28	6.33 ± 0.03 ^A,e^	7.48 ± 0.30 ^A,e^	12.31 ± 2.21 ^B,e^	10.30 ± 0.97 ^C,e^

BC-control brine, 100 % NaCl; BK1-brine with 25% KCl and 75% NaCl; BK2-brine with 50% KCl and 50% NaCl; BK3-brine with 75% KCl and 25% NaCl. ^A–D^ The same superscript capital letters within a row denote no significant differences (*p* > 0.05) between the values obtained for the different trials regarding the control sample according to Tukey’s ANOVA. ^a–e^ The same superscript lowercase letters within a column denotes no significant differences (*p* > 0.05) between values obtained for different days of storage according to Tukey’s ANOVA.

**Table 2 foods-13-03031-t002:** Kinetics of brine properties (pH, total dissolved solids (TDS), conductivity (S), colour parameters (L*, a*, b*)) change over time.

Sample	Brine Property	Kinetic Model	Model Parameters	*R* ^2^	*R* ^2^ _adj_	SEE
BC	pH	Exponential growth	a = 4.7073 ± 0.0230 b = 0.0021 ± 0.0003 1/day	0.8157	0.8016	0.0519
	TDS	Exponential decay	a = 51.1753 ± 0.7624 g/L b = 0.0145 ± 0.0010 1/day	0.9422	0.9378	1.6041
	S	Exponential decay	a = 101.1718 ± 2.0028 mS/cm b = 0.0139 ± 0.0013 1/day	0.8954	0.8874	4.2236
	L*	Exponential decay	a = 98.6112 ± 0.8040 b = 0.0029 ± 0.0005 1/day	0.7300	0.7092	1.7740
	a*	Exponential growth	a = 6.4137 ± 0.2273 b = 0.1163 ± 0.0125 1/day	0.9818	0.9804	0.3243
	b*	Quadratic model	a = −0.426 ± 0.0063 b = 0.0014 ± 0.0002 c = −0.0037 ± 0.0371	0.7933	0.7588	0.0683
BK1	pH	Exponential growth	a = 4.7133 ± 0.0135 b = 0.0021 ± 0.0002 1/day	0.9444	0.9401	0.0306
	TDS	Exponential decay	a = 53.3946 ± 1.1653 g/L b = 0.0137 ± 0.0015 1/day	0.8709	0.8610	2.4597
	S	Exponential decay	a = 106.6447 ± 2.4210 mS/cm b = 0.0132 ± 0.0015 1/day	0.8545	0.8433	5.1194
	L*	Exponential decay	a = 96.9774 ± 3.5023 b = 0.0080 ± 0.0023 1/day	0.4791	0.4390	7.5614
	a*	Exponential growth	a = 7.7399 ± 0.3032 b = 0.1310 ± 0.0171 1/day	0.9725	0.9704	0.4943
	b*	Exponential growth to maximum	a = 0.3479 ± 0.0504 b = 0.1140 ± 0.0495 1/day	0.7109	0.6887	0.0702
BK2	pH	Exponential growth	a = 4.7075 ± 0.0179 b = 0.0034 ± 0.0002 1/day	0.9492	0.9453	0.0407
	TDS	Exponential decay	a = 54.1919 ± 0.9700 g/L b = 0.0130 ± 0.0012 1/day	0.9023	0.8948	2.0528
	S	Exponential decay	a = 107.9383 ± 2.0431 mS/cm b = 0.0122 ± 0.0012 1/day	0.8814	0.8723	4.3365
	L*	Exponential decay	a = 104.9552 ± 4.3242 b = 0.0192 ± 0.0029 1/day	0.7871	0.7707	8.9459
	a*	Exponential growth	a = 0.0799 ± 0.0145 b = 0.0681 ± 0.0073 1/day	0.9101	0.9032	0.0545
	b*	Exponential growth to maximum	a = 24.3896 ± 14.8288 b = 0.0230 ± 0.0018 1/day	0.9008	0.8932	1.3897
BK3	pH	Exponential growth	a = 4.8219 ± 0.0466 b = 0.0029 ± 0.0005 1/day	0.6891	0.6652	0.1057
	TDS	Exponential decay	a = 46.1769 ± 1.1605 g/L b = 0.0157 ± 0.0017 1/day	0.8661	0.8558	2.4311
	S	Exponential decay	a = 93.5068 ± 1.6409 mS/cm b = 0.0177 ± 0.0012 1/day	0.9415	0.9370	3.4129
	L*	Exponential decay	a = 98.4078 ± 5.2847 b = 0.0183 ± 0.0039 1/day	0.6481	0.6210	11.1738
	a*	Exponential growth to maximum	a = 0.9947 ± 0.3790 b = 0.0497 ± 0.0330 1/day	0.7453	0.7257	0.1472
	b*	Exponential growth to maximum	a = 9.8673 ± 1.2965 b = 0.0943 ± 0.0322 1/day	0.8419	0.8297	1.1406

BC-control brine, 100% NaCl; BK1-brine with 25% KCl and 75% NaCl; BK2-brine with 50% KCl and 50% NaCl; BK3-brine with 75% KCl and 25% NaCl.

**Table 3 foods-13-03031-t003:** Mean values for physical characteristics (pH value, titratable acidity (°SH), salt concentration, colour (L*, a*, b*)) of white brined cheese during storage at 4 °C for 28 days.

Variable	Day of Storage	Samples			
		C	K1	K2	K3
pH	0	4.85 ± 0.10 ^A,a^	4.85 ± 0.10 ^A,a^	4.85 ± 0.10 ^A,a^	5.04 ± 0.00 ^A,a^
	7	4.87 ± 0.13 ^A,a^	4.85 ± 0.03 ^A,a^	4.93 ± 0.03 ^A,a^	5.16 ± 0.06 ^B,a^
	14	5.00 ± 0.11 ^A,a^	5.06 ± 0.14 ^A,a^	4.88 ± 0.01 ^A,b^	5.18 ± 0.05 ^A,a^
	21	5.13 ± 0.11 ^A,b^	5.02 ± 0.07 ^A,a^	5.14 ± 0.01 ^A,c^	5.21 ± 0.0 ^A,a^
	28	5.25 ± 0.16 ^A,c^	5.15 ± 0.04 ^A,b^	5.32 ± 0.12 ^A,a^	5.39 ± 0.0 ^A,b^
°SH	0	83.20 ± 0.00 ^A,a^	83.20 ± 0.00 ^A,a^	83.20 ± 0.00 ^A,a^	63.20 ± 0.00 ^B,a^
	7	46.50 ± 7.60 ^A,b^	43.30 ± 9.11 ^A,b^	42.80 ± 15.30 ^A,b^	21.60 ± 1.10 ^B,b^
	14	34.80 ± 5.40 ^A,c^	30.40 ± 9.10 ^A,c^	31.53 ± 10.70 ^A,c^	20.30 ± 0.40 ^A,c^
	21	30.00 ± 4.00 ^A,d^	31.33 ± 1.62 ^A,d^	16.50 ± 1.60 ^A,d^	24.00 ± 0.00 ^A,d^
	28	35.90 ± 1.40 ^A,e^	28.40 ± 0.60 ^A,e^	22.20 ± 1.50 ^A,e^	26.40 ± 0.40 ^A,e^
Salt concentration	7	5.46 ± 0.26 ^A,a^	6.37 ± 0.06 ^B,a^	6.24 ± 0.08 ^C,a^	5.11 ± 0.27 ^A,a^
	14	6.39 ± 0.25 ^A,b^	6.33 ± 0.06 ^A,a^	6.42 ± 0.17 ^A,a^	4.75 ± 0.62 ^A,a^
	21	6.21 ± 0.26 ^A,a^	5.88 ± 0.22 ^A,a^	6.06 ± 0.20 ^A,a^	5.94 ± 0.32 ^A,a^
	28	6.83 ± 0.51 ^A,c^	6.70 ± 0.02 ^A,a^	6.36 ± 0.13 ^A,a^	6.12 ± 0.37 ^A,b^
L*	0	92.45 ± 0.13 ^A,a^	93.45 ± 0.58 ^A,a^	92.01 ± 0.51 ^A,a^	93.16 ± 0.40 ^A,a^
	7	93.52 ± 0.93 ^A,a^	93.52 ± 0.93 ^A,a^	93.52 ± 0.93 ^A,a^	92.46 ± 0.01 ^A,a^
	14	92.13 ± 0.77 ^A,a^	94.35 ± 0.37 ^B,a^	92.46 ± 0.01 ^A,a^	93.40 ± 0.18 ^A,a^
	21	91.49 ± 0.26 ^A,a^	92.17 ± 0.76 ^A,a^	93.40 ± 0.18 ^B,a^	93.27 ± 0.30 ^C,a^
	28	90.37 ± 0.99 ^A,b^	90.61 ± 0.59 ^A,b^	93.27 ± 0.30 ^B,a^	93.16 ± 0.42 ^C,a^
a*	0	−0.49 ± 0.40 ^A,a^	−0.49 ± 0.40 ^A,a^	−0.49 ± 0.40 ^A,a^	−0.49 ± 0.40 ^A,a^
	7	−0.33 ± 0.24 ^A,a^	−0.44 ± 0.22 ^A,a^	−0.50 ± 0.39 ^A,a^	−0.72 ± 0.11 ^A,a^
	14	−0.47 ± 0.43 ^A,a^	−0.96 ± 0.14 ^A,a^	−0.53 ± 0.33 ^A,a^	−0.55 ± 0.11 ^A,a^
	21	−0.50 ± 0.25 ^A,a^	−0.18 ± 0.86 ^A,a^	−0.54 ± 0.23 ^A,a^	−0.21 ± 0.19 ^A,a^
	28	−0.93 ± 0.43 ^A,a^	−0.41 ± 0.64 ^A,a^	−0.35 ± 0.26 ^A,a^	−0.16 ± 0.26 ^A,a^
b*	0	13.68 ± 1.46 ^A,a^	13.68 ± 1.46 ^A,a^	13.68 ± 1.46 ^A,a^	13.68 ± 1.46 ^A,a^
	7	9.76 ± 0.41 ^A,a^	12.73 ± 0.55 ^A,a^	11.80 ± 0.42 ^A,a^	12.58 ± 0.44 ^A,a^
	14	13.11 ± 2.18 ^A,a^	12.73 ± 1.44 ^A,a^	11.68 ± 1.87 ^A,a^	12.53 ± 0.30 ^A,a^
	21	11.71 ± 1.69 ^A,a^	12.35 ± 0.98 ^A,a^	11.08 ± 2.52 ^A,a^	12.40 ± 0.16 ^A,a^
	28	10.45 ± 0.96 ^A,a^	10.74 ± 0.29 ^A,a^	11.13 ± 1.22 ^A,a^	11.85 ± 0.15 ^A,a^

C-control cheese, 100% NaCl; K1-cheese from brine with 25% KCl and 75% NaCl; K2-cheese from brine with 50% KCl and 50% NaCl; K3-cheese from brine with 75% KCl and 25% NaCl. ^A–C^ The same superscript capital letters within a row denote no significant differences (*p* > 0.05) between the values obtained for the different trials regarding the control sample according to Tukey’s ANOVA. ^a–e^ The same superscript lowercase letters within a column denotes no significant differences (*p* > 0.05) between values obtained for different days of storage, according to Tukey’s ANOVA.

**Table 4 foods-13-03031-t004:** Mean values of textural properties characteristics of white brined cheese during storage at 4 °C for 28 days.

Variable	Day of Storage	Samples			
		C	K1	K2	K3
Hardness (N)	7	10.35 ± 3.26 ^A,a^	14.46 ± 4.95 ^A,a^	9.42 ± 1.31 ^A,a^	8.42 ± 1.27 ^A,a^
	14	8.18 ± 3.39 ^A,a^	8.11 ± 1.64 ^A,a^	8.96 ± 3.57 ^A,a^	7.26 ± 1.38 ^A,a^
	21	8.64 ± 3.13 ^A,a^	7.44 ± 0.70 ^A,a^	7.74 ± 0.74 ^A,a^	6.82 ± 4.91 ^A,a^
	28	5.21 ± 0.80 ^A,a^	4.27 ± 0.57 ^B,a^	4.90 ± 2.21 ^A,a^	3.92 ± 0.22 ^A,a^
Adhesive force (N)	7	−0.15 ± 0.04 ^A,a^	−0.17 ± 0.03 ^A,a^	−0.11 ± 0.04 ^A,a^	−0.14 ± 0.04 ^A,a^
	14	−0.11 ± 0.05 ^A,a^	−0.14 ± 0.06 ^A,a^	−0.18 ± 0.08 ^A,a^	−0.18 ± 0.05 ^A,a^
	21	−0.16 ± 0.06 ^A,a^	−0.12 ± 0.02 ^A,a^	−0.15 ± 0.03 ^A,a^	−0.19 ± 0.03 ^A,a^
	28	−0.13 ± 0.04 ^A,a^	−0.09 ± 0.02 ^B,a^	−0.13 ± 0.05 ^A,a^	−0.15 ± 0.05 ^A,a^
Adhesiveness (N mm)	7	0.51 ± 0.11 ^A,a^	0.39 ± 0.29 ^A,a^	0.37 ± 0.15 ^A,a^	0.77 ± 0.25 ^A,a^
	14	0.44 ± 0.45 ^A,a^	0.50 ± 0.10 ^A,a^	0.56 ± 0.23 ^A,a^	0.76 ± 0.13 ^A,a^
	21	0.53 ± 0.07 ^A,a^	0.43 ± 0.09 ^A,a^	0.68 ± 0.18 ^A,a^	0.47 ± 0.20 ^A,a^
	28	0.46 ± 0.19 ^A,a^	0.38 ± 0.05 ^A,a^	0.35 ± 0.02 ^A,a^	0.71 ± 0.25 ^A,a^
Cohesiveness (N/m)	7	0.28 ± 0.05 ^A,a^	0.27 ± 0.01 ^A,a^	0.24 ± 0.01 ^A,a^	0.22 ± 0.04 ^A,a^
	14	0.24 ± 0.03 ^A,a^	0.26 ± 0.02 ^A,a^	0.28 ± 0.05 ^A,a^	0.24 ± 0.02 ^A,a^
	21	0.25 ± 0.02 ^A,a^	0.24 ± 0.04 ^A,a^	0.24 ± 0.04 ^A,a^	0.21 ± 0.02 ^A,a^
	28	0.22 ± 0.02 ^A,a^	0.23 ± 0.03 ^A,a^	0.25 ± 0.02 ^A,a^	0.23 ± 0.02 ^A,a^
Gumminess (N)	7	3.00 ± 1.27 ^A,a^	3.93 ± 1.40 ^A,a^	1.77 ± 0.34 ^A,a^	1.47 ± 0.50 ^A,a^
	14	1.98 ± 0.95 ^A,a^	2.14 ± 0.54 ^A,a^	2.53 ± 1.12 ^A,a^	1.97 ± 0.40 ^A,a^
	21	2.14 ± 0.71 ^A,a^	1.76 ± 0.40 ^A,a^	1.90 ± 0.47 ^A,a^	1.45 ± 1.17 ^A,a^
	28	1.17 ± 0.27 ^A,a^	0.96 ± 0.16 ^B,a^	1.20 ± 0.44 ^A,a^	0.92 ± 0.12 ^A,a^
Postponed elasticity (mm)	7	−3.01 ± 2.28 ^A,a^	−2.02 ± 1.39 ^A,a^	−1.47 ± 0.83 ^A,a^	−3.86 ± 1.19 ^A,a^
	14	−3.70 ± 1.02 ^B,a^	−4.51 ± 0.89 ^B,a^	−4.09 ± 1.25 ^B,a^	−4.88 ± 0.60 ^A,a^
	21	−0.99 ± 1.09 ^A,a^	−3.36 ± 1.75 ^A,a^	−3.87 ± 1.36 ^A,a^	−4.20 ± 1.92 ^A,a^
	28	−5.69 ± 0.98 ^A,a^	−4.75 ± 1.14 ^A,a^	−3.38 ± 1.47 ^A,a^	−5.57 ± 0.38 ^A,a^
Chewiness (N mm)	7	15.92 ± 8.47 ^A,a^	10.82 ± 9.20 ^A,a^	10.41 ± 3.78 ^A,a^	6.49 ± 2.59 ^A,a^
	14	7.89 ± 5.46 ^A,a^	8.49 ± 3.37 ^A,a^	7.71 ± 6.89 ^A,a^	4.54 ± 1.99 ^A,a^
	21	10.73 ± 5.03 ^A,a^	7.18 ± 3.07 ^A,a^	6.98 ± 2.07 ^A,a^	4.11 ± 4.49 ^A,a^
	28	3.00 ± 1.08 ^B,a^	2.13 ± 0.85 ^B,a^	2.92 ± 2.21 ^B,a^	2.09 ± 0.83 ^B,a^
Resistance (N)	7	0.27 ± 0.07 ^A,a^	0.28 ± 0.08 ^A,a^	0.36 ± 0.05 ^A,a^	0.24 ± 0.08 ^A,a^
	14	0.24 ± 0.07 ^A,a^	0.22 ± 0.02 ^A,a^	0.24 ± 0.06 ^A,a^	0.20 ± 0.01 ^A,a^
	21	0.38 ± 0.04 ^A,a^	0.27 ± 0.08 ^A,a^	0.24 ± 0.06 ^A,a^	0.22 ± 0.10 ^A,a^
	28	0.18 ± 0.02 ^A,a^	0.22 ± 0.04 ^A,a^	0.24 ± 0.13 ^A,a^	0.20 ± 0.02 ^A,a^
Breakage (N)	7	10.05 ± 3.35 ^A,a^	7.56 ± 4.72 ^A,a^	7.28 ± 1.26 ^A,a^	5.93 ± 1.15 ^A,a^
	14	7.37 ± 2.80 ^A,a^	5.98 ± 2.87 ^A,a^	5.60 ± 2.31 ^A,a^	5.09 ± 2.18 ^A,a^
	21	8.46 ± 3.05 ^A,a^	5.94 ± 2.67 ^A,a^	5.80 ± 2.22 ^A,a^	6.24 ± 4.95 ^A,a^
	28	4.58 ± 0.57 ^B,a^	4.03 ± 0.62 ^A,a^	4.43 ± 2.46 ^A,a^	3.23 ± 0.65 ^A,a^
Fibrousness (mm)	7	8.04 ± 2.55 ^A,a^	4.72 ± 3.21 ^A,a^	7.35 ± 1.28 ^A,a^	10.22 ± 0.43 ^A,a^
	14	7.43 ± 4.40 ^A,a^	6.20 ± 1.52 ^A,a^	5.67 ± 1.54 ^A,a^	11.74 ± 1.30 ^A,a^
	21	5.61 ± 1.47 ^A,a^	6.80 ± 2.61 ^A,a^	7.91 ± 2.77 ^A,a^	6.12 ± 2.82 ^A,a^
	28	8.69 ± 3.00 ^A,a^	8.15 ± 2.39 ^A,a^	7.68 ± 5.57 ^A,a^	9.63 ± 3.53 ^A,a^

C—control cheese, 100% NaCl; K1—cheese from brine with 25% KCl and 75% NaCl; K2—cheese from brine with 50% KCl and 50% NaCl; K3—cheese from brine with 75% KCl and 25% NaCl. ^A,B^ The same superscript capital letters within a row denote no significant differences (*p* > 0.05) between the values obtained for the different trials regarding the control sample according to Tukey’s ANOVA. ^a^ The same superscript lowercase letters within a column denotes no significant differences (*p* > 0.05) between values obtained for different days of storage according to Tukey’s ANOVA.

**Table 5 foods-13-03031-t005:** Mean values of sensory properties (appearance, colour, consistency, cut, odour, taste) in white brined cheese for the control sample (C), samples brined in 25, 50 and 75% KCl in brine (K1, K2 and K3) at 4 °C over 28 days.

Property	Day of Storage	Samples			
		C	K1	K2	K3
Appearance	7	1.9 ± 0.1 ^A,a^	1.9 ± 0.08 ^A,a^	1.9 ± 0.1 ^A,a^	1.9 ± 0.2 ^A,a^
	14	1.9 ± 0.1 ^A,a^	1.7 ± 0.3 ^A,a^	1.8 ± 0.3 ^A,a^	1.9 ± 0.1 ^A,a^
	21	1.9 ± 0.3 ^A,a^	1.8 ± 0.2 ^A,a^	1.8 ± 0.5 ^A,a^	1.7 ± 0.3 ^A,a^
	28	1.9 ± 0.2 ^A,a^	1.9 ± 0.2 ^A,a^	1.7 ± 0.5 ^A,a^	1.8 ± 0.3 ^A,a^
Colour	7	1.1 ± 0.4 ^A,a^	0.9 ± 0.1 ^A,a^	1.1 ± 0.3 ^A,a^	1.0 ± 0.1 ^A,a^
	14	1.0 ± 0.1 ^A,a^	0.9 ± 0.1 ^A,a^	0.9 ± 0.1 ^A,a^	0.9 ± 0.1 ^A,a^
	21	1.4 ± 1.9 ^A,a^	0.9 ± 0.2 ^B,a^	0.9 ± 0.2 ^B,a^	0.8 ± 0.2 ^B,a^
	28	0.9 ± 0.2 ^A,b^	0.9 ± 0.5 ^A,a^	0.8 ± 0.2 ^A,b^	0.9 ± 0.1 ^A,a^
Consistency	7	1.9 ± 0.3 ^A,a^	1.1 ± 0.3 ^B,a^	2.0 ± 0.1 ^A,a^	1.9 ± 0.2 ^A,a^
	14	2.0 ± 0.0 ^A,a^	0.9 ± 0.1 ^B,a^	2.0 ± 0.1 ^A,a^	1.8 ± 0.3 ^B,a^
	21	1.9 ± 0.1 ^A,a^	0.9 ± 0.2 ^B,a^	1.9 ± 0.2 ^A,a^	1.8 ± 0.4 ^A,a^
	28	1.9 ± 0.2 ^A,a^	0.8 ± 0.2 ^B,b^	1.8 ± 0.4 ^A,a^	1.7 ± 0.6 ^A,a^
Cut	7	3.0 ± 3.8 ^A,a^	2.9 ± 0.2 ^A,a^	3.0 ± 0.1 ^A,a^	2.9 ± 0.1 ^A,a^
	14	3.0 ± 0.0 ^A,a^	3.0 ± 0.1 ^A,a^	3.0 ± 0.1 ^A,a^	2.9 ± 0.3 ^A,a^
	21	3.0 ± 0.0 ^A,a^	2.9 ± 0.2 ^A,a^	2.9 ± 0.2 ^A,a^	3.0 ± 0.1 ^A,a^
	28	2.8 ± 0.4 ^A,a^	2.8 ± 0.5 ^A,a^	2.8 ± 0.4 ^A,a^	2.8 ± 0.3 ^A,a^
Odour	7	2.0 ± 0.1 ^A,a^	1.9 ± 0.2 ^A,a^	1.9 ± 0.2 ^A,a^	1.9 ± 0.2 ^A,a^
	14	1.9 ± 0.3 ^A,a^	1.9 ± 0.1 ^A,a^	1.9 ± 0.1 ^A,a^	1.8 ± 0.2 ^A,a^
	21	2.0 ± 0.0 ^A,a^	1.9 ± 0.1 ^A,a^	1.9 ± 0.2 ^A,a^	1.9 ± 0.1 ^A,a^
	28	2.0 ± 0.0 ^A,a^	1.9 ± 0.3 ^A,a^	1.9 ± 0.3 ^A,a^	2.0 ± 0.1 ^A,a^
Taste	7	9.0 ± 4.0 ^A,b^	9.1 ± 1.1 ^A,a^	9.3 ± 0.8 ^A,a^	7.8 ± 2.7 ^B,a,^
	14	9.5 ± 0.4 ^A,a^	9.3 ± 0.8 ^A,a^	9.3 ± 0.8 ^A,a^	5.8 ± 2.7 ^B,b^
	21	8.9 ± 1.1 ^A,b^	8.7 ± 2.2 ^A,b^	8.1 ± 2.2 ^B,b^	5.0 ± 3.5 ^C,c^
	28	9.2 ± 0.6 ^A,a^	8.4 ± 0.8 ^B,b^	8.3 ± 1.2 ^B,b^	4.6 ± 2.8 ^C,d^

C—control cheese, 100% NaCl; K1—cheese from brine with 25% KCl and 75% NaCl; K2—cheese from brine with 50% KCl and 50% NaCl; K3—cheese from brine with 75% KCl and 25% NaCl. ^A–C^ The same superscript capital letters within a row denote no significant differences (*p* > 0.05) between the values obtained for the different trials regarding the control sample according to Tukey’s ANOVA. ^a–d^ The same superscript lowercase letters within a column denotes no significant differences (*p* > 0.05) between values obtained for different days of storage, according to Tukey’s ANOVA.

**Table 6 foods-13-03031-t006:** Architecture of the ANNs selected for the description of the specific properties of brine and white brined cheese during the storage period.

Sample	Network Structure	Hidden Activation Function	Output Activation Function	Training Perf. Training Error	Test Perf. Test Error	Validation Perf. Validation Error
Brine	MLP 3-9-6 (physical properties)	Tanh	Logistic	0.9722 0.0107	0.9398 0.0184	0.9048 0.0373
			Output variable			
			pH	0.9892	0.9883	0.9782
			S	0.9901	0.9813	0.8861
			TDS	0.9847	0.9777	0.8738
			L*	0.9551	0.8069	0.7651
			a*	0.9626	0.9605	0.9030
			b*	0.9785	0.9650	0.9456
Cheese	MLP 6-4-3 (physical properties)	Tanh	Identity	0.8779 0.0295	0.8744 0.0299	0.8770 0.0468
			Output variable			
			pH	0.8751	0.8370	0.8189
			°SH	0.8574	0.8539	0.8049
			Salt conc.	0.9012	0.8525	0.8425
	MLP 6-7-1 (total colour change)	Tanh	Identity	0.9974 0.0041	0.9687 0.0096	0.9486 0.0112
	MLP 6-8-1 (hardness)	Exponential	Logistic	0.9987 0.0001	0.9985 0.0134	0.8178 0.0175

## Data Availability

The original contributions presented in the study are included in the article, further inquiries can be directed to the corresponding author.

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
