# Peer review of "Utilisation of Potassium Chloride in the Production of White Brined Cheese: Artificial Neural Network Modeling and Kinetic Models for Predicting Brine and Cheese Properties during Storage"

_foods, 2024, doi:10.3390/foods13193031_

Round 1

Reviewer 1 Report

Comments and Suggestions for Authors

The manuscript title was “Utilization of Potassium Chloride in the Production of White Brined Cheese: Artificial Neural Network Modeling and Kinetic Models for Predicting Brine and Cheese Properties During Storage”. The research was about the potassium chloride replace the sodium chloride in the brined cheese to reduce the high blood pressure. The reaearch was meaningful.

The specific revision advice was as follow.

1.     I think the significance or contribution of artificial neural network modeling and kinetic models in this research should be introduced in the introduction.

2.     There should be spaces between numbers and units.

3.     “Potassium chloride (KCl) was used to replace NaCl in the brine in a ratio of 25%, 50% and 75% (w/V)” What is the basis for the selection of different concentrations?

4.     What temperature and humidity is the cheese stored at?

5.     Does the total amount of potassium chloride have an impact on human health, within what range should it be appropriate, and is the relevant research sufficient?Please cite relevant literature for elaboration.

Author Response

The manuscript title was “Utilization of Potassium Chloride in the Production of White Brined Cheese: Artificial Neural Network Modeling and Kinetic Models for Predicting Brine and Cheese Properties During Storage”. The research was about the potassium chloride replace the sodium chloride in the brined cheese to reduce the high blood pressure. The research was meaningful.

Response: Thank you very much for your comment!

Point-by-point response to Comments and Suggestions for Authors

Comments 1: I think the significance or contribution of artificial neural network modeling and kinetic models in this research should be introduced in the introduction.

Response 1: Thank you for pointing this out. We agree with this comment. We have updated the introduction part. Lines 81-95 (red color)

Comments 2: There should be spaces between numbers and units.

Response 2: We have put spaces between units and numbers.

Comments 3: “Potassium chloride (KCl) was used to replace NaCl in the brine in a ratio of 25%, 50% and 75% (w/V)” What is the basis for the selection of different concentrations?

Response 3: Thank you for pointing this out. After reviewing the available literature, we found that the most common substitutions are between 25 % and 50 %, in one work even 70 %. We could not find any examples of higher substitutions, only lower percentages. We decided to try a 75% substitution of NaCl. We were interested in how such a high amount of substitute salt would affect the texture, preservation and taste of the product. Please take a look at the literature we provided.

1.      Renata Golin Bueno Costa, Rayane Campos Alves, Adriano Gomes da Cruz, Denise Sobral, Vanessa Aglaê Martins Teodoro, Luiz Carlos Gonçalves Costa Junior, Junio César Jacinto de Paula, Taynan Barroso Landin, Elisângela Michele Miguel (2018) Manufacture of reduced-sodium Coalho cheese by partial replacement of NaCl with KCl, International Dairy Journal, 87, 37-43, https://doi.org/10.1016/j.idairyj.2018.07.012.

2.      Emilie Thibaudeau, Denis Roy, Daniel St-Gelais (2015) Production of brine-salted Mozzarella cheese with different ratios of NaCl/KCl. International Dairy Journal, 40,

54-61. https://doi.org/10.1016/j.idairyj.2014.07.013.

3.      Hugo L.A. Silva, Celso F. Balthazar, Erick A. Esmerino, Roberto P.C. Neto, Ramon S. Rocha, Jeremias Moraes, Rodrigo N. Cavalcanti, Robson M. Franco, Maria Inês B. Tavares, Jânio S. Santos, Daniel Granato, Renata G.B. Costa, Mônica Q. Freitas, Márcia C. Silva, Renata S.L. Raices, C. Senaka Ranadheera, Filomena Nazzaro, Amir M. Mortazavian, Adriano G. Cruz (2018) Partial substitution of NaCl by KCl and addition of flavor enhancers on probiotic Prato cheese: A study covering manufacturing, ripening and storage time. Food Chemistry, 248, 192-200. https://doi.org/10.1016/j.foodchem.2017.12.064.

Comments 4: What temperature and humidity is the cheese stored at?

Response 4: The cheese was kept in the refrigerator during the 28-day storage period. The temperature was 4 °C and we did not measure the humidity, as the cheeses were stored in glass containers immersed in brine.

Comments 5: Does the total amount of potassium chloride have an impact on human health, within what range should it be appropriate, and is the relevant research sufficient? Please cite relevant literature for elaboration.

Response 5: Thank you for your question. Potassium chloride (KCl) is generally recognized as safe (GRAS status) and has been extensively studied for its effects on human health. When substituting KCl for sodium chloride (NaCl) in foods, it is important to consider the total potassium intake in the diet, as excessive consumption may have health consequences for individuals with certain conditions such as chronic kidney disease. The World Health Organization (WHO) and other health authorities recommend a potassium intake of at least 3.51 mg per day for an average adult to counteract the negative effects of sodium and support cardiovascular health (WHO, 2012). However, the upper limit for potassium intake is less clearly defined, as the body is able to efficiently excrete excess amounts in healthy individuals (Aburto et al., 2013). While there is no generally established upper limit for potassium intake from food for the general population, the Institute of Medicine (now the National Academy of Medicine) suggests an upper limit of 470 mg/day (IOM, 2005). It is recommended that potassium intake should be balanced and not exceed these guidelines, especially in populations at risk of hyperkalemia. Excessive potassium intake is usually a problem for people with impaired kidney function, where hyperkalemia can pose a health risk (Weaver, 2013). A study by Geleijnse et al (2010) supports the use of KCl as a partial replacement for NaCl and shows that potassium-enriched salt can lower blood pressure in healthy individuals without adverse effects. According to the National Institute of Health, the recommended daily intake of KCl for adults is about 3.5 g per day.

The highest concentration of KCl we used was 75% KCl in the brine, with 25% NaCl. Since each brine was made with 10 % salt, this means that 1 liter of brine contained 75 g KCl and 25 g NaCl. Cheese and brine were filled into glass containers at a ratio of 1:4, i.e. 100 g cheese and 400 ml brine. The results show that the salt content in the samples with 75 % KCl is between 4 % and 6 %, which corresponds to a salt content of 4 - 6 g in the cheese. As the Mohr method for salt determination measures the concentration of chloride ions, it can be assumed that between 3 and 4.5 g of KCl are present in the cheese. However, this is only a guess, as the method cannot distinguish which chloride ions come from sodium and which from potassium. Based on these results and the current recommendations, it can be concluded that even this highest substitution concentration (75 %) is harmless to health, with the exception of people with kidney problems. In such cases, however, the exact amount of potassium contained in the product should be indicated on the product label.

References:

1.      Aburto, N. J., Hanson, S., Gutierrez, H., Hooper, L., Elliott, P., Cappuccio, F. P. (2013) Effect of increased potassium intake on cardiovascular risk factors and disease: systematic review and meta-analyses. BMJ, 346, f1378. DOI: 10.1136/bmj.f1378

2.      Geleijnse, J. M., Witteman, J. C., Bak, A. A., den Breeijen, J. H., Grobbee, D. E. (2010). Reduction in blood pressure with a low sodium, high potassium, high magnesium salt in older subjects with mild to moderate hypertension. BMJ, 315(7116), 542-544.

DOI: 10.1136/bmj.309.6952.436

3.      Institute of Medicine (IOM). (2005). Dietary Reference Intakes for Water, Potassium, Sodium, Chloride, and Sulfate. Washington, DC: The National Academies Press.

4.      Weaver, C. M. (2013). Potassium and health. Advances in Nutrition, 4(3), 368S–377S.

5.      World Health Organization (WHO). (2012). Guideline: Potassium Intake for Adults and Children. Geneva: World Health Organization.

Reviewer 2 Report

Comments and Suggestions for Authors

This manuscript explores the effects of partially replacing NaCl with KCl at different levels in white brined cheese and compares the physicochemical properties of the brine and cheese before and after substitution. PCA is used to identify the key factors influencing cheese quality, and ANNs are applied to predict the properties of both the brine and the cheese. The study shows a certain degree of innovation and practical application value, but the following aspects require revision to improve the overall quality of the manuscript.

(1)   Please standardize abbreviations throughout the text. If an abbreviation has already been introduced earlier (with the full form and abbreviation provided), the abbreviation can be used in the following parts of the text.

(2)   Line 18, “with brine cheese” needs to be revised as “with brined cheese”. There are other points in the text that need to be modified, please check.

(3)   Line 27, It is suggested that modifying the “texture” to “physicochemical properties” can more accurately reflect the research content of the manuscript.

(4)   Line 33, please modify the reference format according to the requirement of the journal.

(5)   Line 49, please add the unit to 19.

(6)   Line 76, please check if “by” is appropriate.

(7)   Line 99, please add the unit to 3.

(8)   Line 225, is this section “Results” or “Results and Discussion”? As there is no separate discussion section behind it.

(9)   Line 271-276, 308-309, 370-376, 427-428, please move these notes to the bottom of the table.

(10)  Line 311, please add the number if it is a title.

(11)  Line 348-350, KCl (74) has a higher molecular weight than NaCl (58), so please check the explanation.

(12)  L394-396, there was no the correlation analysis in the manuscript, and it is biased to draw such a conclusion.

(13)  Line 426, is it a special feta cheese formula? Or white brined cheese? In Table 4, the data of adhesive force were not marked the results of ANOVA. Moreover, the uppercase and lowercase letters indicate meaning without annotation.

(14)  Line 456, the distinction between colour indicators for colour and taste is difficult to differentiate.

(15)  Line 472-477, these sentences are the same as the ones above.

(16)  Line 576 and 598, please add the linear equations and determination coefficients to the Figure 4 and Figure 5.

(17)  Line 600, the author has previously conducted research on replacing NaCl with calcium salts (calcium lactate and calcium citrate). Compare with the results of the current study, which method is more suitable for substitution?

Comments on the Quality of English Language

English language is good, but some content needs to be written more concisely.

Author Response

This manuscript explores the effects of partially replacing NaCl with KCl at different levels in white brined cheese and compares the physicochemical properties of the brine and cheese before and after substitution. PCA is used to identify the key factors influencing cheese quality, and ANNs are applied to predict the properties of both the brine and the cheese. The study shows a certain degree of innovation and practical application value, but the following aspects require revision to improve the overall quality of the manuscript.

Response: Thank you very much for your comment!

Point-by-point response to Comments and Suggestions for Authors

Comments 1: Please standardize abbreviations throughout the text. If an abbreviation has already been introduced earlier (with the full form and abbreviation provided), the abbreviation can be used in the following parts of the text.

Response 1: Thank you for pointing this out. The abbreviations have been standardized in the text.

Comments 2:  Line 18, “with brine cheese” needs to be revised as “with brined cheese”. There are other points in the text that need to be modified, please check.

Response 2: Thank you for recognising our mistake. Throughout the text (in red), “white brine cheese” has been changed to “white brine cheese”

Comments 3:  Line 27, It is suggested that modifying the “texture” to “physicochemical properties” can more accurately reflect the research content of the manuscript.

Response 3: Thank you for pointing this out. It has been changed.

Comments 4: Line 33, please modify the reference format according to the requirement of the journal.

Response 4: We have changed it.

Comments 5:  Line 49, please add the unit to 19.

Response 5: The unit (%) was added.

Comments 6: Line 76, please check if “by” is appropriate.

Response 6: Thank you for recognising our mistake. "by" has been changed to "from".

Comments 7: Line 99, please add the unit to 3.

Response 7: The unit (%) was added.

Comments 8:  Line 225, is this section “Results” or “Results and Discussion”? As there is no separate discussion section behind it

Response 8: According to the instructions to the authors, they can choose whether the results should be published together with the discussion or separately. We have opted for a joint presentation of results and discussion and we have corrected the title in the manuscript.

Comments 9: Line 271-276, 308-309, 370-376, 427-428, please move these notes to the bottom of the table.

Response 9: Thank you for taking note. In the above lines, the notes have been moved to the two ends of the tables. The change has been written in red.

Comments 10: is 1/day unit?

Response 10: 1/day represents the rate constant for the exponential growth or exponential decay of a parameter (e.g. pH, TDS or conductivity) over time. The unit 1/day indicates that the rate is expressed per day, i.e. it indicates how fast the parameter changes (increases or decreases) per day. In an exponential decay model, the rate constant (e.g. b = 0.0157 1/day) indicates how fast the total dissolved solids (TDS) decreases per day. Similarly, in an exponential growth model for the pH value, the rate constant (e.g. b = 0.0021 1/day) indicates the daily rate of increase of the pH value.

Comments 11: Line 311, please add the number if it is a title.

Response 11: A number has been added to the title and other title numbers have been changed.

Comments 12:  Line 348-350, KCl (74) has a higher molecular weight than NaCl (58), so please check the explanation.

Response 12: Thank you very much for your comment! Yes, we made a mistake, which we have now corrected. Please see in the text (red colour).

Comments 13: L394-396, there was no the correlation analysis in the manuscript, and it is biased to draw such a conclusion.

Response 13: Thank you for your comment. It is true that the PCA we performed uses the correlations (or covariance’s) between the variables as part of its process to transform the data, but it is not a correlation analysis per se, so we delete this conclusion.

Comments 14: Line 426, is it a special feta cheese formula? Or white brined cheese? In Table 4, the data of adhesive force were not marked the results of ANOVA. Moreover, the uppercase and lowercase letters indicate meaning without annotation.

Response 14: No, that was a mistake. We have corrected Table 4 according to your instructions. Thank you for your comment.

Comments 15:  Line 456, the distinction between colour indicators for colour and taste is difficult to differentiate.

Response 15: Thank you for your comment. Instead of Figure 1, we have opted for a table with all the data so that the results are more transparent and visible.

Comments 16: Line 472-477, these sentences are the same as the ones above.

Response 16: Thank you for the comment. Yes, there were repetitions of the sentences. We have deleted the repetition of the sentences.

Comments 17: Line 576 and 598, please add the linear equations and determination coefficients to the Figure 4 and Figure 5.

Response 17: Linear equations and determination coefficients have been added.

Comments 18:  Line 600, the author has previously conducted research on replacing NaCl with calcium salts (calcium lactate and calcium citrate). Compare with the results of the current study, which method is more suitable for substitution?

Response 18: Thank you for your interest in our work. In conclusion part we have added two sentences regarding to our previous work.

Reviewer 3 Report

Comments and Suggestions for Authors

This manuscript aimed to use potassium chloride in the production of white brined cheese. I have some suggestions to improve its quality.   

ABSTRACT

- " In addition, the  study investigates the use of ANN (Artificial Neural Network) models to predict certain brine and cheese properties and the result showed that ANN models can be used to predict changes in brine and cheese properties over time."- Please, present this result more clear.

INTRODUCTION

- L. 33 - Verify citation form.

- L. 49 - 19%?

- L. 42-80 - This paragraph is too lengthy and it present repeated information. Please, re-vise it. Furthermore, the innovation of the manuscript should be clear stated, as several papers deal with substitution of the NaCl for KCl. What is the novelty here?

- The introduction section could provide some information about artificial neural network, which was used in the manuscript.

MATERIAL AND METHODS

- L. 100- How was it pasteurized? Heat exchanger? Water bath?

- How was the storage time chosen? Is 28 days the shelf life of commercial white brined cheeses?

- Please, include the approval of the ethics committee for the sensory analysis.

RESULTS

- Include the p-values for differences or not (p < 0.05 or p > 0.05) for all results.

- L. 319-320 - Why did K3 show a higher pH just after processing?

- L. 329-332 - Please, include the results of proteolysis to confirm this explanation.

- L. 389 - Why? Need a better explanation.

- L. 399-401 - How? Please, explain.

- L. 436 - Feta cheese?

- Figure 1 - Please, replace this figure by a Table with the sensory results and statistical analysis.

Author Response

This manuscript aimed to use potassium chloride in the production of white brined cheese. I have some suggestions to improve its quality.  

Response: Thank you very much for your suggestions.

Point-by-point response to Comments and Suggestions for Authors

Comments 1:  " In addition, the  study investigates the use of ANN (Artificial Neural Network) models to predict certain brine and cheese properties and the result showed that ANN models can be used to predict changes in brine and cheese properties over time."- Please, present this result more clear.

Response 1: Thank you for pointing this out. We have written clearer results regarding ANNs.

Comments 2:   L. 33 - Verify citation form.

Response 2: Thank you for recognising our mistake. We have changed it.

Comments 3:  L. 49 - 19%?

Response 3: The unit (%) was added.

Comments 4: L. 42-80 - This paragraph is too lengthy and it present repeated information. Please, re-vise it. Furthermore, the innovation of the manuscript should be clear stated, as several papers deal with substitution of the NaCl for KCl. What is the novelty here?

Response 4: Thank you for your comment, we have revised the introduction part and added some new data about the novelty of our work.

Comments 5:  - The introduction section could provide some information about artificial neural network, which was used in the manuscript.

Response 5: Thank you for your comment. We have provided information on ANN, red colour text.

Comments 6: - L. 100- How was it pasteurized? Heat exchanger? Water bath?

Response 6: Pasteurization was carried out in a water bath. We have now written it in the manuscript.

Comments 7: - How was the storage time chosen? Is 28 days the shelf life of commercial white brined cheeses?

Response 7: According to our legislation, white brined cheese must be left in brine for at least 2 weeks before it is sold. We have therefore decided to carry out the analyses after twice this time and white brined cheese is often placed on the market after 28 days of storage.

Comments 8:  - Please, include the approval of the ethics committee for the sensory analysis.

Response 8: We have sent to the Foods editorial team declarations of consent from for our sensory analyst. They also ask us for approval.

Comments 9: Include the p-values for differences or not (p < 0.05 or p > 0.05) for all results.

Response 9: We have given p-values for all results.

Comments 10: - L. 319-320 - Why did K3 show a higher pH just after processing?

Response 10: We assume that the higher pH value is due to the highest KCl concentration in the salt solution. However, since none of the ions in these solutions can change the concentration of H⁺ or OH ions in the water, NaCl and KCl solutions should have essentially the same pH. When the experiments were repeated (they were repeated three times), the salt solution had a higher pH each time when the highest amount of KCl was in the salt solution. It is possible that some interactions between white brine cheese and brine occur immediately.

Comments 11:  L. 329-332 - Please, include the results of proteolysis to confirm this explanation.

Response 11: We have not performed proteolysis analyses. This statement comes from a group of authors who performed a proteolysis analysis. We have cited them.

1.      Kamleh R, Olabi A, Toufeili I, Daroub H, Younis T, Ajib R. The effect of partial substitution of NaCl with KCl on the physico-691 chemical, microbiological and sensory properties of Akkawi cheese. J Sci Food Agric. 2015 Jul;95(9):1940-8.

Comments 12: L. 389 - Why? Need a better explanation.

Response 12: Thank you for your comment and your interest in this work. The literature states that cheese tends to become softer in texture as the KCl content in the brine increases. This is because KCl has a different ionic effect on the cheese proteins and the moisture content. KCl does not firm the cheese as much as NaCl, resulting in a softer, more delicate texture. In summary, using a brine with a higher concentration of KCl compared to NaCl in cheese making can result in a softer cheese texture. Furthermore, NaCl determines water activity and thus controls microbial growth, enzymatic activity and biochemical changes during cheese ripening, while at the same time developing the desired flavour and aroma. Together with pH and calcium content, NaCl has a significant influence on the hydration and aggregation of paracasein, which in turn affects the ability of the casein matrix to bind water, the tendency of the matrix to syneresis, its rheological and textural properties.

Guinee T. P. (2004) Salting and the role of salt in cheese, International Journal of Dairy Technology 57 (2-3): 99-109.

Comments 13: - L. 399-401 - How? Please, explain.

Response 13: This explanation can be associate with the explanation above. When only NaCl is used, the cheese often has a firmer and more compact texture due to the stronger interaction between casein proteins and the reduced water content (as NaCl reduces water activity). This contributes to the formation of a firm network and makes the cheese firmer.

The use of KCl results in less solidification of the protein structures. This means that the casein protein structure does not form as firmly, resulting in a softer and more delicate cheese texture. KCl can also cause the cheese to retain more moisture, which contributes to a softer texture. Here are references which can confirm this response:

1.      Wang, L.F.; Zhang, X.W. Effects of potassium chloride on the quality characteristics of reduced-sodium cheese. J. Dairy Sci. 2012, 95, 7348-7355.

2.      Guinee, T. P. (2004). "Cheese and Fermented Milk Products." In: Handbook of Food Science, Technology, and Engineering, edited by Y. H. Hui, CRC Press.

3.      Fox, P. F., & McSweeney, P. L. H. (2004). "Dairy Chemistry and Biochemistry." Springer.

4.      Wang, L. F., & Zhang, X. W. (2012). "Effects of potassium chloride on the quality characteristics of reduced-sodium cheese." Journal of Dairy Science, 95(12), 7348-7355.

5.      Keenan, T. W., & McDermott, J. (2007). "Salt and cheese: The role of sodium and potassium chloride in cheese manufacture." International Dairy Journal, 17(5), 493-503.

6.      Jelen, P., & Winger, R. J. (2011). "Cheese Technology." In: Dairy Science and Technology Handbook, edited by L. P. Tompkins, CRC Press.

Comments 14:  - L. 436 - Feta cheese?

Response 14: Sorry, it is a mistake. White brined cheese.

Comments 15: Figure 1 - Please, replace this figure by a Table with the sensory results and statistical analysis..

Response 15: Thank you for your suggestion. We have replaced Figure 1 with Table 5 and added a statistical analysis.